# Balancing efficiency and clinical quality in dialysis centers: Insights from a nationwide DEA study in Taiwan

**Shu-Chuan Jennifer Yeh**[1]*, **Wen Chun Wang**[2], **Andrew Pruett**[3], **Hsueh-Chih Chou**[4]

**1** Institute of Health Care Management & Department of Business Management, National Sun Yat-sen University, Kaohsiung, Taiwan, **2** Department of Forensic Psychiatry, Kaohsiung Municipal Kai-Syuan Psychiatric Hospital, Kaohsiung, Taiwan, **3** Department of Business Management, National Sun Yat-sen University, Kaohsiung, Taiwan, **4** Department of Nursing, Kaohsiung Veterans General Hospital, Kaohsiung, Taiwan

* syeh@faculty.nsysu.edu.tw

## Abstract

### Objectives

The increasing prevalence of end-stage renal disease (ESRD), especially in aging populations, presents significant challenges for healthcare systems. Dialysis centers must navigate growing demands for cost efficiency while maintaining high-quality care. This study aimed to evaluate the relationship between operational efficiency and clinical quality in dialysis centers, using a systems-based performance assessment framework.

### Methods

We analyzed 578 dialysis centers in Taiwan using Data Envelopment Analysis (DEA) to estimate operational efficiency. Clinical quality was assessed using outcome indicators including mortality rate, dialysis adequacy (urea reduction ratio [URR], Kt/V), serum albumin and hemoglobin levels, calcium-phosphate (Ca×P) product, and cardiothoracic ratio. Multiple regression analyses were conducted to examine associations between efficiency scores, clinical outcomes, and organizational characteristics, including chain affiliation and ownership type.

### Results

Higher efficiency scores were significantly associated with lower URR, Kt/V, and Ca×P values, suggesting potential trade-offs between operational efficiency and clinical quality. Centers affiliated with chains generally reported better clinical outcomes. For-profit centers exhibited higher URR, Kt/V, and albumin levels, as well as lower Ca×P values, compared to their non-profit counterparts.

**Data availability statement:** The minimal data set underlying the findings of this study cannot be made publicly available due to ethical and legal restrictions related to the confidentiality of participating dialysis centers and the terms of data use approved by the data-holding institution. The data are restricted by the Taiwan Nephrology Nursing Association (TNNA). Researchers who meet the criteria for access to confidential data may request the minimal data set by contacting the TNNA through the following non-author institutional point of contact: Chia-Hua Wu, Secretary General Taiwan Nephrology Nursing Association Email: tnnanew@ms51.hinet.net Address: 6F-1, No. 9, Lane 155, Nanjing West Road, Datong District, Taipei City 103, Taiwan Website: https://www.tnna.org.tw/home/clinic_evaluate.asp.

**Funding:** This study was supported by the National Science and Technology Council, Taiwan (MOST 109-2410-H-110-017-MY3). The funder had no role in the study design, data collection and analysis, decision to publish, or preparation of the manuscript.

**Competing interests:** No authors have competing interests.

## Conclusions

Operational efficiency in dialysis centers may come at the cost of certain clinical outcomes. However, organizational characteristics such as chain affiliation and for-profit ownership are linked to better quality indicators. These findings highlight the value of DEA as a tool for system-level performance evaluation and inform strategies to optimize dialysis care delivery.

## Introduction

Chronic kidney disease (CKD) affects over 800 million individuals worldwide, with end-stage renal disease (ESRD) requiring lifelong dialysis [1]. The burden of ESRD is particularly evident in regions like Taiwan, the U.S., and Singapore, where incidence rates remain among the highest globally [2]. As dialysis places a significant strain on healthcare resources, optimizing efficiency while maintaining high-quality care is a critical policy concern. Research on dialysis center efficiency has yielded mixed findings, with studies suggesting that facility size, ownership structure, and market competition influence performance [3]. Understanding these dynamics is essential to improving dialysis service delivery and patient outcomes.

Efficiency in dialysis care has far-reaching financial and clinical implications. In Taiwan, where dialysis costs consume a substantial portion of the national healthcare budget, providers must balance growing demand with financial sustainability [4]. Prior research suggests that factors such as chain affiliation and ownership type may impact operational efficiency and care quality. For example, studies indicate that nonprofit dialysis centers often exhibit lower technical efficiency than their for-profit counterparts, though competition can drive improvements [3]. By assessing the interplay between efficiency, organizational structure, and clinical quality, this study aims to inform policy decisions that enhance dialysis care value.

Quality of care in dialysis is measured through key indicators such as dialysis adequacy (Kt/V and URR), anemia management (hemoglobin levels), bone metabolism (calcium-phosphate balance), and hemodynamic stability [5]. These clinical markers directly influence patient survival and well-being, underscoring the need for performance benchmarking. To evaluate operational efficiency in dialysis centers, we employed Data Envelopment Analysis (DEA), a non-parametric method used to assess the relative efficiency of decision-making units with multiple inputs and outputs [6]. DEA is particularly suited for healthcare settings because it allows simultaneous consideration of multiple resource inputs (e.g., staff, equipment) and outputs (e.g., clinical quality indicators) without assuming a specific functional form [7]. In this study, we applied the BCC model to calculate efficiency scores for each center, where a score of 1 indicates a fully efficient center relative to its peers [8]. These efficiency scores provide an objective measure of operational efficiency, which we subsequently related to clinical-quality indicators using regression analyses to identify best practices in dialysis management [7,9]. However, while efficiency improvements streamline operations, excessive focus on cost reduction may compromise

individualized patient care [10,11]. Prior studies suggest that highly efficient hospitals may experience higher mortality rates, highlighting the delicate balance between operational efficiency and patient-centered care [12].

This study examines dialysis center performance using secondary data, focusing on the relationship between efficiency, organizational type, and care quality. This research considers how dialysis providers manage financial and operational constraints to enhance patient outcomes. By evaluating differences between for-profit and nonprofit centers, as well as chain-affiliated versus independent facilities, this study provides insights into how healthcare organizations navigate external dependencies. Findings from this research will contribute to policy recommendations aimed at improving the sustainability and quality of dialysis care.

## Materials and methods

### Research design and data

This study is a retrospective, cross-sectional study of the relative efficiency of dialysis facilities for the year 2017–2018 and clinical performance indicators at the organizational level. The study dataset was developed by using 2017–2018 accreditation data from the Taiwan Nephrology Nurse Association. The data contains information on the characteristics of dialysis centers, devices, dialyzers, and staffing. Because hemodialysis is the predominant dialysis treatment in Taiwan, this study focused exclusively on hemodialysis (HD) data, which included all 578 HD centers nationwide. These centers therefore represent the entire population of hemodialysis facilities, while peritoneal dialysis centers were excluded from the analysis. The unit of analysis was the individual dialysis center, and no identifiable patient-level or detailed organizational information was available to the researchers.

The research has been approved by the Human Research Ethics Committee at National Cheng Kong University (NCK-UHREC) (Approval No. NCKU HREC-E-108-299-2). The data were accessed for research purposes on January 11, 2022.

### Measurement

**Dependent variables.** Several healthcare quality measures for dialysis patients served as the dependent variables. These were chosen due to their direct relation to quality of hemodialysis. These measures are accepted quality standards as defined by Taiwan's National Health Insurance. First, annual mortality for each dialysis center was calculated as the number of deaths divided by total patient-years at risk, with each patient contributing proportionally to the time they received hemodialysis during the year, and expressed as a percentage. Second, to evaluate the adequacy of hemodialysis, two indicators would be selected: urea reduction ratio (URR, $\geqq 0.65$), Kt/V ($\geqq 1.2$), and serum albumin (Albumin, $\geqq 3.5$ g/dl with BCG or $\geqq 3.0$ g/dl with BCP). Third, for monitoring anemia, the index would be hemoglobin (Hb, $\geqq 8.5$ g/dl). Fourth, in assessing bone metabolism, we employed the calcium-phosphate product (Ca × P) values, with a threshold of ≥60. A value exceeding 60 suggests an unfavorable outcome. Finally, for assessing patients' hemodynamic status, we chose the cardiothoracic ratio (CTR $\leqq 0.55$) as another dependent variable, where higher values indicate worse outcomes. The cardiothoracic ratio (CTR) is vital for renal dialysis patients as it aids in evaluating heart size, early detection of cardiovascular issues, and determining fluid status. This information is crucial for managing their elevated risk of heart-related complications [13]. The clinical quality variables in our study—such as dialysis adequacy (Kt/V and URR), hemoglobin levels for anemia management, calcium–phosphate balance, and other biochemical markers—are grounded in widely recognized national and international quality frameworks for dialysis care. These indicators are emphasized in the Kidney Disease Outcomes Quality Initiative (KDOQI) and associated quality measure initiatives, which outline core measures for dialysis adequacy and management of anemia and mineral metabolism that are linked to patient outcomes and survival [14]. The unit of analysis in this study was the dialysis center. Although the clinical indicators originate from individual patient records, all measures used in the analysis were aggregated at the facility level and reported as center-level summary statistics (e.g., mean URR, mean Kt/V, proportion meeting

hemoglobin targets). Therefore, all variables included in the regression models reflect organizational-level outcomes rather than individual-level data.

**Independent variables.** *Efficiency score*: We used the Data Envelopment Analysis (DEA) technique to measure the relative technical efficiency scores of each dialysis center. Each dialysis center is called Decision Making Units (DMUs) in DEA. This study used a variable return to scale (VRS, BCC) DEA model in examining the technical efficiency of dialysis facilities, assuming that the average productivity at the most productive scale size may not be attainable for the other scale sizes at which a given dialysis center may be operating [13]. DEA provided efficiency scores by taking account of both multiple inputs and outputs [9,15]. For dialysis centers, the selected input variables included the number of beds, number of nephrologists, number of nurses, number of dialysis technicians, number of other clinical staff full-time equivalents (FTEs), and the number of patients occupying a dialysis station per day. These inputs were chosen because they are closely related to care quality and have been supported in previous studies, indicating their suitability for inclusion in our model [3,16,17]. The output variable was the number of dialysis patients. Efficiency scores were assessed on a scale ranging from 0 to 1. A score of 1 denoted a relatively efficient unit, whereas a score below 1 indicated inefficiency.

*Chain-/nonchain-affiliated*: In this research, dialysis centers belonging to the chain medical institution or health care systems might be classified as chain-affiliated units. Chain-affiliated were coded as 1 and non-chain affiliated were coded as 0, as reference.

*For-profits/nonprofits*: Public and foundation hospitals were defined as non-profit hospitals (coded as 1) in Taiwan. On the other side, corporations and private hospitals were defined as for-profit hospitals (coded as 0).

**Control variables.** For dialysis patients, older age is associated with lower levels of physical functioning [18], and higher mortality [19]. For dialysis quality, females had better Kt/V than males, and males had better quality in hemoglobin than females [20]. Patients with more comorbidities were at increased risk of post-dialysis mortality [21]. Preexisting comorbidities are associated with the mortality rate as well as the pre-dialysis adverse events in incident dialysis patients. Hence, in this study, we incorporated gender (percentage of females), age (average age of each center, which was defined as real years), and comorbidities (average numbers of patients' comorbidities) as control variables.

## Data analysis

To gain a comprehensive understanding of our sample, we first conducted descriptive analyses. For categorical variables, such as ownership and profit status, we calculated frequencies and percentages. For continuous variables, including age, comorbidity, efficiency scores, and others, we examined the mean and standard deviation.

Next, we explored relationships among all variables using Spearman's rank correlation, a nonparametric alternative to Pearson's correlation. This method allowed us to assess both the strength and direction of associations between variables.

Third, we conducted multiple regression analyses to test our main hypotheses. Each healthcare outcome—mortality rate, URR, KT/V, albumin, hemoglobin, Ca×P, and CTR—was analyzed separately as a dependent variable. Independent variables included efficiency, chain affiliation, and profit status, while control variables were incorporated to account for their influence on the dependent variables and reduce potential confounding effects. Model fit was evaluated using adjusted R-square, F-values, and p-values.

Finally, to assess the robustness of the findings, sensitivity analyses were performed by excluding dialysis centers with extreme scale characteristics. Specifically, centers in the top and bottom 5% of total patient visits and those in the top and bottom 5% of number of beds were excluded. Multiple regression analyses were then re-run for all seven clinical outcome variables.

## Results

### Sample characteristics

This study examines 578 hemodialysis centers, of which 250 (43.3%) are chain-affiliated, while 328 (56.7%) operate independently. The majority—426 centers (73.7%)—are for-profit, with the remaining 152 (26.3%) classified as nonprofit. On average, patients in these centers are 65.52 years old and have 2.83 comorbidities. The average efficiency score is 0.95, ranging from 0.32 to 1.0. Facility-level clinical outcomes indicated a mean mortality rate of 0.06 (SD = 0.04), mean URR of 0.75 (SD = 0.02), mean Kt/V of 1.41 (SD = 0.09), mean albumin of 3.88 g/dL (SD = 0.18), mean hemoglobin of 10.41 g/dL (SD = 0.59), mean Ca × P of 45.12 (SD = 4.01), and mean cardiothoracic ratio of 0.51 (SD = 0.03). A detailed breakdown of these characteristics is presented in Table 1.

### Univariate analysis

Table 2 presents a summary of Spearman's rank correlation coefficient between the variables. Efficiency score was significantly correlated with Mortality rate ($r = -.086$, $p < .05$), URR ($r = -.128$, $p < .01$) and KT/V ($r = -.127$, $p < .01$). Chain-affiliated was significantly correlated with Mortality rate ($r = .118$, $p < .01$), URR ($r = .087$, $p < .05$), KT/V ($r = .087$, $p < .05$) and Hemoglobin ($r = .095$, $p < .05$). For-profits was significantly correlated with mortality rate ($r = -.098$, $p < .05$), URR ($r = .089$, $p < .05$), Albumin ($r = .130$, $p < .01$) and CTR ($r = -.121$, $p < .01$).

### Multivariate analyses

We performed multiple regression to test the hypothesis. The efficiency score ($\beta = -.016$, SE = .05, $p < .001$), affiliation with a chain ($\beta = .008$, SE = .002, $p < .01$), and for-profit status ($\beta = .008$, SE = .002, $p < .001$) were identified as predictors for URR. The higher KT/V were associated with less efficiency score ($\beta = -.064$, $SE = .023$, $p < .01$), chain

**Table 1. Descriptive characteristics.**

|  | Min | Max | Mean | SD | Frequency | Percentage (%) |
|---|---|---|---|---|---|---|
| Female% | .03 | .80 | .48 | .07 | -- | -- |
| Age | 52.24 | 75.91 | 65.52 | 2.59 | -- | -- |
| Comorbidity | .00 | 10.88 | 2.83 | 1.71 | -- | -- |
| Efficiency score | .32 | 1.00 | .95 | .16 | -- | -- |
| Chain |  |  |  |  |  |  |
| Yes | -- | -- | -- | -- | 250 | 43.3 |
| No | -- | -- | -- | -- | 328 | 56.7 |
| For-profit |  |  |  |  |  |  |
| Yes | -- | -- | -- | -- | 426 | 73.7 |
| No | -- | -- | -- | -- | 152 | 26.3 |
| Mortality rate | .00 | .31 | .06 | .04 | -- | -- |
| URR | .66 | .81 | .75 | .02 | -- | -- |
| KT/V | 1.11 | 1.84 | 1.41 | .09 | -- | -- |
| Albumin | 3.21 | 4.64 | 3.88 | .18 | -- | -- |
| Hemoglobin | .27 | 12.02 | 10.41 | .59 | -- | -- |
| Ca × P | 16.22 | 58.23 | 45.12 | 4.01 | -- | -- |
| CTR | .03 | .60 | .51 | .03 | -- | -- |

Note. N = 578; URR = urea reduction ratio; KT/V = K – dialyzer clearance of urea, T – dialysis time, V – volume of distribution of urea; Ca × P = calcium-phosphate product; CTR = cardiothoracic ratio.

**Table 2. Spearman's rank correlation coefficient.**

|  | 1 | 2 | 3 | 4 | 5 | 6 | 7 | 8 | 9 | 10 | 11 | 12 |
|---|---|---|---|---|---|---|---|---|---|---|---|---|
| 1. Female% | -- |  |  |  |  |  |  |  |  |  |  |  |
| 2. Age | .123** | -- |  |  |  |  |  |  |  |  |  |  |
| 3. Comorbidity | −0.02 | 0.05 | -- |  |  |  |  |  |  |  |  |  |
| 4. Efficiency score | −0.07 | −0.03 | 0.00 | -- |  |  |  |  |  |  |  |  |
| 5. Chain | −.166** | 0.03 | −0.07 | −0.06 | -- |  |  |  |  |  |  |  |
| 6. For-profit | .137** | −.104* | 0.06 | .175** | −.422** | -- |  |  |  |  |  |  |
| 7. Mortality rate | 0.01 | .191** | .215** | −.086* | .118** | −.098* | -- |  |  |  |  |  |
| 8. URR | .221** | 0.07 | 0.05 | −.128** | .087* | .089* | 0.02 | -- |  |  |  |  |
| 9. KT/V | .233** | 0.07 | 0.04 | −.127** | .087* | 0.06 | 0.02 | .974** | -- |  |  |  |
| 10. Albumin | −0.05 | −.230** | 0.03 | 0.03 | −0.02 | .130** | −0.03 | 0.00 | −0.01 | -- |  |  |
| 11. Hemoglobin | −.093* | −.095* | .105* | 0.02 | .095* | −0.07 | 0.05 | .094* | .095* | .215** | -- |  |
| 12. Ca×P | 0.01 | −.281** | −0.02 | 0.05 | −0.07 | −0.08 | −0.05 | −.177** | −.158** | .204** | 0.07 | -- |
| 13. CTR | .126** | .192** | 0.07 | −0.06 | 0.03 | −.121** | .100* | −0.01 | −0.02 | −.143** | −.115** | −0.06 |

*Note.* \*\*\*$p < .001$ \*\*$p < .01$ \*$p < .05$. URR = urea reduction ratio; KT/V = K – dialyzer clearance of urea, T – dialysis time, V – volume of distribution of urea; Ca×P = calcium-phosphate product; CTR = cardiothoracic ratio.

affiliated ($\beta = .029$, $SE = .008$, $p < .001$) and for-profit ($\beta = .024$, $SE = .009$, $p < .05$). Centers with a for-profit status were associated with albumin outcomes ($\beta = .090$, $SE = .019$, $p < .001$). The variables that predicted calcium-phosphate product were efficiency score ($\beta = 2.361$, $SE = 1.001$, $p < .05$), chain affiliated ($\beta = −.835$, $SE = .357$, $p < .05$) and for-profit ($\beta = −1.388$, $SE = .413$, $p < .001$). Mortality rate, hemoglobin, and CTR had no significant relationships with efficiency and organization types.

Multicollinearity was assessed using variance inflation factors (VIF), tolerance statistics, and condition index diagnostics, following established methodological guidelines [22–24]. All VIF values were below commonly accepted thresholds (VIF < 5, or under more liberal criteria, < 10), and all tolerance values exceeded 0.20, indicating no evidence of substantial pairwise multicollinearity.

Condition index diagnostics were further evaluated in conjunction with variance-decomposition proportions [22]. Although one condition index exceeded 30 in the model including the for-profit indicator, only a single variable (percentage of female patients) exhibited a high variance-decomposition proportion (0.96) for that dimension, while no other variables showed substantial variance proportions. According to established criteria, problematic multicollinearity is indicated only when two or more variables exhibit large variance proportions associated with the same high condition index. Therefore, these diagnostics do not suggest the presence of meaningful multicollinearity in the model. VIF, tolerance, and condition index values are also presented in Table 3.

## Results of sensitivity analyses

Overall, the sensitivity analyses produced results that were consistent with the primary analyses. The direction and magnitude of the associations between operational efficiency, clinical outcomes, and organizational characteristics remained largely unchanged across all seven dependent variables. In the model with Kt/V as the dependent variable, exclusion of centers with extreme numbers of beds resulted in a loss of statistical significance, although the estimated effect size and direction were similar to those observed in the full-sample analysis. These findings suggest that the main conclusions are robust and not driven by dialysis centers with extreme patient volume or bed capacity.

Table 3. Results of multiple regression analyses with multicollinearity diagnostics.

| Variable | Mortality rate | | URR | | KT/V | | Albumin | | Hemoglobin | | Ca×P | | CTR | | Toler-ance | VIF | Condition Index |
|---|---|---|---|---|---|---|---|---|---|---|---|---|---|---|---|---|---|
| | β | SE | β | SE | β | SE | β | SE | β | SE | β | SE | β | SE | | | |
| Female% | −.004 | .025 | .070*** | .013 | .310*** | .055 | −.002 | .107 | −.254 | .365 | 3.969 | 2.390 | .042* | .018 | .932 | 1.073 | 3.045 |
| Age | .003*** | .001 | .000 | .000 | .001 | .001 | −.015*** | .003 | −.029** | .010 | −.450*** | .063 | .001** | .000 | .959 | 1.042 | 4.976 |
| Comorbidity | .006*** | .001 | .001 | .001 | .004 | .002 | .008 | .004 | .039** | .014 | −.028 | .094 | .001 | .001 | .984 | 1.016 | 6.522 |
| Efficiency score | −.020 | .011 | −.016** | .005 | −.064** | .023 | −.002 | .045 | −.072 | .153 | 2.361* | 1.001 | −.002 | .007 | .953 | 1.050 | 15.210 |
| Chain | .007 | .004 | .008*** | .002 | .029*** | .008 | .027 | .016 | .097 | .055 | −.835* | .357 | .001 | .003 | .809 | 1.236 | 24.273 |
| For-profit | −.002 | .004 | .008*** | .002 | .024* | .009 | .090*** | .019 | −.065 | .063 | −1.388** | .413 | −.006 | .003 | .767 | 1.304 | 88.726 |
| *R²* | .099 | | .100 | | .094 | | .099 | | .039 | | .101 | | .037 | | | | |
| *Adjusted R²* | .090 | | .091 | | .084 | | .090 | | .028 | | .092 | | .027 | | | | |
| *F-value* | 10.477*** | | 10.599*** | | 9.834*** | | 10.480*** | | 3.821** | | 10.719**** | | 3.667** | | | | |

Note. $N = 578$; ***$p < .001$ **$p < .01$ *$p < .05$; URR = urea reduction ratio; KT/V = K – dialyzer clearance of urea, T – dialysis time, V – volume of distribution of urea; Ca×P = calcium-phosphate product; CTR = cardiothoracic ratio.

## Discussion

The finding that higher efficiency scores were associated with lower URR and Kt/V values suggests that facilities operating more efficiently may be doing so by reducing treatment time or resource intensity, potentially compromising dialysis adequacy. Both URR and Kt/V are key indicators of solute clearance, with clinical guidelines recommending a URR of at least 65% and a single-pool Kt/V (spKt/V) target of approximately 1.4 per treatment to ensure sufficient urea removal and prevent adverse outcomes. The negative association observed in this study indicates that greater operational efficiency may coincide with reduced clearance performance, raising concerns that cost-saving strategies–such as shorter dialysis sessions, lower staffing ratios, or streamlined processes–might inadvertently limit treatment effectiveness. This interpretation is further supported by the positive association between efficiency and the calcium–phosphate (Ca×P) product, where higher values exceed the recommended threshold of $<55\,mg^2/dL^2$ and indicate suboptimal mineral metabolism management. Elevated Ca×P levels are clinically concerning because they are associated with vascular calcification and increased cardiovascular risk. Taken together, these patterns highlight a potential tension between operational efficiency and clinical quality, underscoring the need for balanced performance strategies that safeguard both throughput and patient outcomes.

Ownership type further influenced clinical performance. For-profit centers demonstrated higher URR and Kt/V values, higher albumin levels, and lower Ca×P compared with nonprofit centers, suggesting that investor-owned facilities may allocate resources toward maintaining dialysis adequacy and nutritional monitoring. However, our dataset did not include information on the cost of dialysis sessions across ownership types. Including cost data—such as per-session charges, out-of-pocket expenses, or reimbursement structures—would have provided a more complete perspective on how financial models affect accessibility, equity, and quality. Existing literature underscores the heterogeneous effects of ownership on dialysis care. Lin et al. reported that nephrologist ownership was associated with lower ESA use without worsening anemia control [25], whereas Thamer et al. found that for-profit dialysis facilities administered higher ESA doses than nonprofit facilities, consistent with reimbursement-driven incentives rather than clinical need [26]. Similarly, Eggleston et al. showed that for-profit hospitals respond to financial incentives with variable quality outcomes [27]. While some studies suggest quality advantages for nonprofit hospitals [28], meta-analyses have linked for-profit ownership to higher mortality in certain settings [29,30], and European evidence shows no consistent benefits from for-profit expansion [31]. Overall, these mixed findings highlight the need for context-sensitive interpretation of profit-oriented ownership models in dialysis care.

Chain affiliation and investor ownership may also confer operational advantages. These centers could invest in standardized care pathways, staff training, or real-time monitoring systems that help preserve dialysis adequacy even while pursuing efficiency gains. Importantly, efficiency was not significantly associated with mortality in our models, indicating that reductions in URR/Kt/V and increases in Ca × P do not necessarily translate into measurable short-term mortality differences at the facility level. Nevertheless, these process-oriented deficits–such as lower clearance performance and higher Ca × P–remain clinically meaningful, as prior studies have linked them to higher risks of hospitalization, vascular complications, and long-term morbidity.

Overall, these results underscore the need for policymakers and dialysis center managers to monitor process and biochemical quality indicators, not only cost or throughput metrics, when implementing efficiency-focused strategies. Future research should examine specific organizational practices, including treatment duration, staffing mix, and phosphate-management protocols, and incorporate longitudinal patient-level analyses to determine whether the observed trade-offs persist over time and ultimately affect long-term patient outcomes.

## Implications and future directions

This study offers several implications for policy and practice. Policymakers should consider regulatory frameworks that align efficiency incentives with patient-centered outcomes, ensuring that financial performance does not compromise care quality. In Taiwan, for example, accreditation systems could incorporate minimum quality thresholds–such as URR, Kt/V, and Ca × P control–to ensure that highly efficient centers continue to meet essential clinical standards. Reimbursement models under the National Health Insurance (NHI) could also integrate blended or pay-for-performance approaches that reward not only efficient resource use but also consistently strong clinical outcomes, such as stable hemoglobin levels or improved dialysis adequacy. Additionally, health insurance incentives that promote adherence to evidence-based protocols may help counterbalance potential efficiency–quality trade-offs. Collaboration between independent and chain-affiliated centers could further promote equity in resource distribution, dissemination of standardized protocols, and overall performance improvement across the dialysis sector. From a systems operations perspective, administrators can apply these findings to refine dialysis center management. Investments in health IT systems, workforce development, and AI-enabled tools can enhance both efficiency and care delivery. Flexible operational strategies, such as modular staffing and data-informed workflow redesign, can further optimize service capacity without eroding care standards. Future research should integrate multi-level data—linking patient characteristics, organizational structures, and financial parameters—to better understand the causal mechanisms underlying these associations. Longitudinal studies are especially needed to assess how organizational strategies and incentive structures influence clinical outcomes over time. Embracing a systems-thinking approach will be essential to sustaining high-quality, efficient dialysis care in an era of increasing demand and resource constraints.

## Limitations and strengths

This study has several limitations, but also notable strengths. First, its cross-sectional and ecological design limits the ability to draw causal inferences between efficiency indicators and clinical outcomes. Because the analyses were conducted using aggregated, facility-level data rather than individual-level information, the observed associations may not accurately reflect relationships that occur among individual patients. This reliance on group-level measures also introduces the potential for ecological fallacy, and the cross-sectional nature of the data prevents establishing temporal ordering, making causal interpretations inappropriate.

Second, the model's explanatory power was modest. The $R^2$ values generally ranged from 9% to 10% and were particularly low for hemoglobin (3.9%) and cardiothoracic ratio (3.7%), indicating that only a small proportion of the variance in these outcomes is explained by the included variables. As clinical outcomes are shaped by multiple patient-, provider-, and system-level factors, these low $R^2$ values are not unexpected; however, they underscore the need for caution in

interpreting the findings. Future research may benefit from alternative analytic approaches—such as hierarchical modeling or interaction terms—to capture more complex relationships among patient load, staffing patterns, and facility size.

Third, although the analyses adjusted for key patient characteristics such as age, gender, and comorbidity burden, other relevant sources of confounding could not be accounted for. Information on medication use, nutritional status, and adherence behaviors—all of which influence albumin and calcium–phosphate control—was unavailable. Additionally, facility-level contextual variables such as geographic characteristics, local market competition, and neighborhood socio-economic environment were not included. These unmeasured factors may have affected both operational efficiency and clinical outcomes.

Fourth, the national dataset lacked several important structural and process quality indicators at the facility level, including cleanliness and environmental hygiene, infection control practices, water treatment quality assurance, and the availability of multidisciplinary support services such as dietitians and social workers. These components represent essential dimensions of dialysis center quality and may influence both organizational efficiency and clinical performance. The absence of these measures limits the comprehensiveness of our performance assessment. Future studies incorporating these structural and process indicators would provide a more holistic evaluation of dialysis center performance.

Despite these limitations, the study has several important strengths. Its primary strength lies in its comprehensive, system-level analysis of dialysis center efficiency, incorporating both operational performance and care quality dimensions. By employing Data Envelopment Analysis (DEA) alongside regression modeling, the study provides a rigorous framework to evaluate facility-level efficiency and identify potential trade-offs with clinical outcomes. In addition, comparing independent and chain-affiliated centers, as well as for-profit versus nonprofit ownership, contributes actionable knowledge for system-level improvement strategies and offers valuable insights for policymakers and healthcare managers seeking to optimize resource allocation while maintaining care quality.

## Conclusion

This paper makes a significant contribution by broadening the scope of dialysis efficiency literature to include dialysis centers. We identified associations between efficiency scores, chain affiliation, profit status, and the clinical quality of dialysis centers. Our findings revealed that efficient centers exhibited lower outcomes in terms of URR, KT/V, and Ca×P values. In contrast, chain-affiliated dialysis centers were more likely to access necessary resources, leading to better outcomes. Additionally, for-profit centers demonstrated proficiency in reallocating resources to enhance the quality of care. In conclusion, these findings highlight the importance of organizational factors in determining the performance and quality of dialysis centers, suggesting avenues for future research and improvement in care delivery.

## Author contributions

**Conceptualization:** Shu-Chuan Jennifer Yeh.

**Data curation:** Hsueh-Chih Chou.

**Formal analysis:** Wen Chun Wang, Andrew Pruett.

**Methodology:** Shu-Chuan Jennifer Yeh, Wen Chun Wang.

**Supervision:** Shu-Chuan Jennifer Yeh.

**Writing – original draft:** Shu-Chuan Jennifer Yeh.

**Writing – review & editing:** Shu-Chuan Jennifer Yeh, Wen Chun Wang, Andrew Pruett, Hsueh-Chih Chou.

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
