## [Decision Letter · Decision Letter 0]

11 Nov 2025

Dear Dr. Yeh,

Thank you for submitting your manuscript to PLOS ONE. After careful consideration, we feel that it has merit but does not fully meet PLOS ONE’s publication criteria as it currently stands. Therefore, we invite you to submit a revised version of the manuscript that addresses the points raised during the review process.

We look forward to receiving your revised manuscript.

Kind regards,

Edward Zimbudzi

Academic Editor

PLOS ONE

Journal Requirements:

2. As you are reporting a retrospective study of medical records or archived samples, please ensure that you have discussed whether all data were fully anonymized before you accessed them and/or whether the IRB or ethics committee waived the requirement for informed consent. If patients provided informed written consent to have data from their medical records used in research, please include this information.

“National Science and Technology Council, Taiwan”

4. In the online submission form you indicate that your data is not available for proprietary reasons and have provided a contact point for accessing this data. Please note that your current contact point is a co-author on this manuscript. According to our Data Policy, the contact point must not be an author on the manuscript and must be an institutional contact, ideally not an individual. Please revise your data statement to a non-author institutional point of contact, such as a data access or ethics committee, and send this to us via return email. Please also include contact information for the third party organization, and please include the full citation of where the data can be found.

Reviewer's Responses to Questions

**Comments to the Author**

1. Is the manuscript technically sound, and do the data support the conclusions?

Reviewer #1: Yes

Reviewer #2: Partly

2. Has the statistical analysis been performed appropriately and rigorously?

Reviewer #1: I Don't Know

Reviewer #2: I Don't Know

3. Have the authors made all data underlying the findings in their manuscript fully available?

Reviewer #1: Yes

Reviewer #2: No

4. Is the manuscript presented in an intelligible fashion and written in standard English?

Reviewer #1: Yes

Reviewer #2: Yes

Reviewer #1: Dear authors,

Your manuscript presents a relevant and timely investigation into the interplay between operational efficiency and clinical quality in dialysis centers. The use of Data Envelopment Analysis (DEA) applied to a nationally representative dataset in Taiwan contributes meaningful insights to the health systems literature—particularly in the context of resource-constrained environments. The study is well-structured, and the narrative is clearly articulated.

The findings have potential implications for healthcare policy, quality management, and the economic evaluation of chronic disease care services. However, certain methodological and analytical aspects warrant further clarification or enhancement to strengthen the robustness and applicability of the study's conclusions.

Below is a detailed assessment based on key elements of research design and scientific merit.

Study Design and Level of Evidence

Your study is a retrospective cross-sectional analysis based on aggregated facility-level data. In the hierarchy of evidence, this design provides hypothesis-generating insights but does not support causal inference.

Although DEA is a well-established method for efficiency analysis, its results are sensitive to model specification, input-output selection, and contextual constraints. These limitations are partially acknowledged but would benefit from further discussion.

Recommendation:

Please emphasize in the limitations section that, due to its cross-sectional and ecological nature, the study does not allow for causal conclusions regarding efficiency and clinical outcomes.

Clarity of Research Question and Objectives

The research objective is clearly defined and clinically relevant. You address a critical question: How do organizational structures and ownership models influence the balance between operational efficiency and clinical quality in dialysis care?

This question is particularly pertinent in aging populations and settings with universal health coverage.

Strengths:

Well-formulated aim.

Clear alignment with global health policy discussions.

Addresses a current gap in the efficiency-effectiveness literature.

Methodology and Internal Validity

You employed a DEA model with variable returns to scale (VRS/BCC) to estimate technical efficiency, followed by multiple regression analyses. The methodology is generally sound but requires additional details and refinement.

Points for consideration:

Selection Criteria: The manuscript does not fully explain the inclusion/exclusion criteria for the 578 centers. Was this the entire population or a subset?

DEA Model Justification: While VRS is appropriate, it would be helpful to provide a rationale for the specific input and output variables chosen, including any theoretical or empirical justification.

Input-Output Correlation: Have you tested for multicollinearity among input/output variables? This is particularly important in DEA, where redundancy can distort the frontier.

Recommendation:

Include a more detailed description of the DEA specification process, including variable selection criteria, orientation (input vs. output), and possible sensitivity analyses.

Statistical Analysis

Your statistical approach is appropriate for an observational study. The use of Spearman’s correlation and multiple regression models is sound, and the results are clearly presented.

However:

The adjusted R² values are relatively low across models, indicating limited explanatory power.

The risk of omitted variable bias remains, given that patient-level covariates (e.g., duration of dialysis, nutritional status, adherence, SES) are not available.

Recommendation:

Discuss the implications of the low R² values more explicitly and consider exploring alternative modeling strategies or interaction terms in future research.

Results Interpretation and Discussion

The discussion section is balanced and thoughtful. You correctly highlight the trade-offs between operational efficiency and certain clinical quality indicators (e.g., URR, Kt/V, Ca × P). Your analysis of ownership models and chain affiliation is well-integrated with the existing literature.

Strengths:

Strong comparative analysis between chain-affiliated vs. independent centers.

Inclusion of economic theory perspectives enriches the interpretation.

Citations are current and relevant.

Suggestion for Improvement:

Expand the discussion on policy implications. For example:

How might your findings inform accreditation systems, reimbursement models, or health insurance incentives in Taiwan or elsewhere?

Risk of Bias and Conflicts of Interest

You have appropriately declared no competing interests, and the study protocol was reviewed and approved by the National Cheng Kung University Research Ethics Committee. Data anonymity is preserved.

However, the following risks of bias should be discussed more clearly:

Selection bias: Particularly if not all dialysis centers were included.

Ecological bias: Outcomes are attributed at the center level, yet clinical variables may vary significantly at the patient level.

Confounding: Control variables are limited and do not fully account for center-level heterogeneity.

Recommendation:

Include a subsection explicitly addressing these sources of bias and their potential influence on findings.

References and Scientific Contribution

Your reference list is comprehensive and well-curated, including seminal works on DEA, dialysis quality measures, and health systems performance.

The study adds value by:

Applying a nationwide dataset in a real-world context.

Linking organizational models to measurable clinical indicators.

Reinforcing the need to integrate efficiency and quality metrics in healthcare evaluations.

Overall Recommendation

Recommendation: Minor Revisions

Your manuscript is suitable for publication after minor revisions that enhance methodological transparency, deepen the discussion on policy implications, and better acknowledge the study's design limitations.

Reviewer #2: My detailed comments and suggestions are provided below.

1. Conceptual clarity: Clinical quality vs. clinical effectiveness

While the study title is “Balancing Efficiency and Clinical Quality in Dialysis Centers,” the objective mentioned in the abstract, as well as in the main text, states that the study is to examine the relationship between operational efficiency and clinical effectiveness. Subsequently, in the methods section, the term clinical quality reappears. Although these concepts are related, they are not interchangeable and are measured differently. Based on the variables and analyses presented, it appears that the study primarily examines clinical quality rather than clinical effectiveness. I recommend using clinical quality consistently throughout the manuscript and ensuring conceptual alignment across the title, objectives, and methods.

2. Introduction of the DEA approach: The Data Envelopment Analysis (DEA) approach should be introduced and briefly explained either at the end of the Introduction section or as part of the method section. Since DEA is a relatively specialised method, readers may not be familiar with it. Please describe the rationale for using DEA, the model applied and how it is suited to assess efficiency in dialysis centres. Currently, the limited information provided under the subheading efficiency score is unclear.

3. Clarification of unit of analysis and variable selection: In Line 115, you indicate that the unit of analysis is the dialysis centre, yet the variables described are predominantly individual-level (patient-level) indicators. If the analysis is indeed at the facility level, it would be important to include facility-level parameters that reflect structural and process aspects of care quality- such as cleanliness, infection control measures, water quality assurance, and availability of support services.

4. Please clarify how variables under efficiency and quality were conceptualised and developed. Were these guided by any national or international quality frameworks or standards for dialysis centres? If so, please cite and describe these sources in the Methods section.

5. The operational definition of mortality (Line 121) is not clear. Please revise that.

6. Need for descriptive overview before analysis: Before the univariate and correlation analyses, it would strengthen the paper to provide a descriptive summary of key variables, including key characteristics of dialysis centres, patient profiles, and outcomes. This would help readers, especially policymakers and practitioners, to better understand the functioning and variability among dialysis centres in Taiwan. Currently, the results focus mainly on statistical correlations, which yield limited new insights without context. A more structured descriptive comparison across different types of dialysis centres could enhance the originality and policy relevance of the findings.

7. Discussion: The Discussion section should be more closely aligned with the findings presented. At present, it appears somewhat speculative, for example, lines 215–221, where efficiency is discussed without sufficient grounding in the results. As noted earlier, including a descriptive overview of variables in the Results section would help substantiate these interpretations. Similarly, in Lines 224–236, you discuss about chain-affiliated centres; however, the cost factor is not included in the results. A comparison of the cost of dialysis sessions across different types of centres would have added a valuable perspective to this discussion, particularly regarding accessibility and equity of care.

8. Please also include a “Strengths and Limitations” subsection following the Discussion. In lines 213 and 214, you mention that variables such as patient education, dialysis duration, and comorbidities, which significantly affect outcomes, were unavailable. Since these are very important components of quality of care, not including those in your analysis is an inherent limitation of the study- these should be explicitly acknowledged in the manuscript.

**Do you want your identity to be public for this peer review?** For information about this choice, including consent withdrawal, please see our Privacy Policy

Reviewer #1: **Yes:** André Luis C Ramalho, MD PhD - Faculty of Medicine - University of Porto, Portugal

Reviewer #2: **Yes:** Maya Annie Elias

---

## [Author Response · Author response to Decision Letter 1]

22 Dec 2025

Please refer to the response-to-reviewers file for detailed responses to each comment.

---

## [Decision Letter · Decision Letter 1]

3 Feb 2026

Balancing Efficiency and Clinical Quality in Dialysis Centers: Insights from a Nationwide DEA Study in Taiwan

PONE-D-25-38110R1

Dear Dr. Yeh,

We’re pleased to inform you that your manuscript has been judged scientifically suitable for publication and will be formally accepted for publication once it meets all outstanding technical requirements.

Kind regards,

Edward Zimbudzi

Academic Editor

PLOS One

Additional Editor Comments (optional):

Reviewers' comments:

Reviewer's Responses to Questions

**Comments to the Author**

Reviewer #1: All comments have been addressed

Reviewer #2: All comments have been addressed

2. Is the manuscript technically sound, and do the data support the conclusions?

Reviewer #1: Yes

Reviewer #2: Yes

3. Has the statistical analysis been performed appropriately and rigorously?

Reviewer #1: Yes

Reviewer #2: I Don't Know

4. Have the authors made all data underlying the findings in their manuscript fully available?

Reviewer #1: Yes

Reviewer #2: Yes

5. Is the manuscript presented in an intelligible fashion and written in standard English?

Reviewer #1: Yes

Reviewer #2: Yes

Reviewer #1: The authors have adequately addressed all comments raised during the previous round of review. The revised manuscript demonstrates substantial improvements in methodological transparency, clarity of reporting, and depth of discussion.

In particular, the authors now clearly acknowledge the cross-sectional and ecological nature of the study and appropriately emphasize the limitations regarding causal inference and potential ecological bias. The description and justification of the DEA model specification, including the rationale for input and output selection, have been strengthened and are now well supported by the literature.

The statistical analyses are rigorous and clearly reported. The inclusion of multicollinearity diagnostics (VIF, tolerance, and condition index) and sensitivity analyses enhances the robustness and credibility of the findings. The discussion of the relatively low R² values is transparent and appropriately contextualized, avoiding overinterpretation of the results.

Importantly, the expanded discussion on policy and managerial implications adds translational value to the study and increases its relevance for health system decision-makers. The manuscript is well written, logically structured, and the conclusions are appropriately aligned with the data presented.

Overall, this manuscript is now technically sound and suitable for publication. I have no further substantive comments.

Reviewer #2: (No Response)

**Do you want your identity to be public for this peer review?** For information about this choice, including consent withdrawal, please see our Privacy Policy

Reviewer #1: **Yes:** Andre Luis C Ramalho, PhD

Reviewer #2: **Yes:** Maya Annie Elias

---

## [Editor Report · Acceptance letter]

PONE-D-25-38110R1

PLOS One

Dear Dr. Yeh,

I'm pleased to inform you that your manuscript has been deemed suitable for publication in PLOS One. Congratulations! Your manuscript is now being handed over to our production team.

Kind regards,

on behalf of

Dr. Edward Zimbudzi

Academic Editor

PLOS One